# Impact of the COVID-19 Pandemic on the Evolution of Prevalence and Patterns of Cannabis Use among First-Year University Students in Spain—UniHcos Project

**DOI:** 10.3390/ijerph191811577

**Published:** 2022-09-14

**Authors:** Lorena Botella-Juan, Carmen Amezcua-Prieto, María M. Morales-Suarez-Varela, Ramona Mateos-Campos, Carlos Ayán-Pérez, Antonio José Molina, Rocío Ortiz-Moncada, Susana Redondo-Martín, Juan Alguacil, Gemma Blázquez-Abellán, Miguel Delgado-Rodríguez, Jessica Alonso-Molero, Tania Fernández-Villa

**Affiliations:** 1Department of Biomedical Sciences, Area of Preventive Medicine and Public Health, Faculty of Health Sciences, University of León, 24071 León, Spain; 2Consortium for Biomedical Research in Epidemiology & Public Health (CIBER Epidemiología y Salud Pública-CIBERESP), 28029 Madrid, Spain; 3Department of Preventive Medicine and Public Health, University of Granada, 18016 Granada, Spain; 4Instituto de Investigación Biosanitaria (ibs.Granada), 18014 Granada, Spain; 5Department of Preventive Medicine and Public Health, Food Sciences, Toxicology and Legal Medicine, University of Valencia, 46100 Burjassot, Spain; 6Area of Preventive Medicine and Public Health, University of Salamanca, 37007 Salamanca, Spain; 7Well-Move Research Group, Department of Special Didactics, University of Vigo, 36310 Vigo, Spain; 8The Research Group in Gene-Environment and Health Interactions (GIIGAS), Institute of Biomedicine (IBIOMED), University of León, 24071 León, Spain; 9Area of Preventive Medicine and Public Health, Food and Nutrition Research Group, University of Alicante, 03550 Alicante, Spain; 10Department of Pathological Anatomy, Microbiology and Preventive Medicine and Public Health, School of Medicine, Universidad de Valladolid, 47005 Valladolid, Spain; 11Comisionado Regional para la Droga, Junta de Castilla y León, 47009 Valladolid, Spain; 12Centre for Research on Natural Resources, Health, and Environment (RENSMA), University of Huelva, 21071 Huelva, Spain; 13Department of Medical Sciences, Faculty of Pharmacy, University of Castilla-La Mancha, 02008 Albacete, Spain; 14Department of Health Sciences, University of Jaén, 23071 Jaén, Spain; 15Departamento de Medicina Preventiva y Salud Pública, Universidad de Cantabria-IDIVAL, 39008 Santander, Spain

**Keywords:** cannabis, marijuana use, students, university, COVID-19, quarantine, public health

## Abstract

Among university students there has been evidence that the COVID-19 pandemic increased their psychological distress, exacerbated by social restrictions. The main objective of this study was to explore the impact of the COVID-19 pandemic on the prevalence and patterns of cannabis use among university students, in contrast to previous trends since 2012. Data from 10,522 first-year university students (73.3% female, M_age_ 19 (SD = 1.6)) from eleven Spanish universities collected between 2012 and May 2022 was analysed. Prevalences of cannabis use and their differences by sex were studied, as well as changes in patterns of use and its use for coping during the pandemic. It was found that during lockdown, all prevalence rates of cannabis use decreased in both sexes, showing no statistically significant differences and increasing again in the new normal period in both. Among regular cannabis users, 79.7% reported maintaining or increasing their cannabis use during the pandemic, and of these, half reported using cannabis to cope. Moreover, cannabis use in the usual household increased during the lockdown. These results show that although the overall prevalence of cannabis use was reduced during the lockdown, regular users tended to maintain or increase cannabis use. This could imply two different patterns of use among students, one social and occasional versus the other regular, providing new lines of research for prevention and the implementation of social policies.

## 1. Introduction

The COVID-19 pandemic has had a major impact on society in different aspects such as socio-economic factors and mental health [1]. In this sense, multiple studies have already assessed the changes in mental health generated in different populations and countries, as well as the risk and protective factors that influence this response, showing a very high impact in general. However, there are clear differences depending on the studied population characteristics (age group, gender, country) and the previous coping capacity and resilience, and also to the lockdown conditions [2,3,4]. Among university students, it has been observed that the impact on their mental health was even greater [5,6], possibly since this is a population with high rates of psychological distress [7]. Furthermore, the beginning of the university years has been associated with peaks of increased substance use such as alcohol and cannabis [8,9]. This fact is related to changes in lifestyle and social circles, along with less family supervision and increased independence [10].

In Spain, cannabis is the most widely used illegal sale substance, ranking in recent studies as one of the European countries with the highest prevalence of cannabis use [11,12,13]. Spanish legislation on cannabis use is quite restrictive, with medical use highly restricted and recreational use allowed in the personal and private sphere while sale and growing for sale as well as use in public spaces is banned [14]. Compared to other EU countries, the legislation is similar to that which has been applied in Germany, France, and Italy, and as in these countries, proposals for less restrictive regulations for cannabis content are beginning to be put forward, in line with the changes set out in decision 63/17 of the Commission on Narcotic Drugs [15]. According to the latest 2021 report on alcohol, tobacco, and illegal drugs carried out by the OEDA (Spanish Observatory on Drugs and Addictions) based on the EDADES surveys (survey on alcohol and other drugs in Spain), the highest prevalence rates of cannabis use are found among the population aged 15–24 years, with a prevalence rate of 22.1% in the past 12 months and 15.9% in the past 30 days, with males showing higher prevalence rates in all the frequencies of use [16]. However, some authors have reported that prevalence rates of cannabis use may be higher among university students than in the general population [17,18].

Within the motivations among this population for cannabis use, it is appropriate to draw a distinction between the motivations of occasional and regular users [19]. Initiation into cannabis use is usually motivated by a variety of reasons such as promoting social integration due to peer pressure, experimenting with new adulthood, and increasing disinhibition to cope with social anxiety [20,21,22]. Thus, occasional use usually takes place in the social sphere, at gatherings and parties. About the influence of the COVID-19 pandemic, the large number of social and mobility restrictions imposed in Spain during the lockdown may have influenced cannabis use in the social context [23]. However, the evolution towards more regular cannabis use and a more intensive use pattern is related to other motivations such as the desire to increase positive feelings, routine, or coping with emotional stress and negative emotions [19,24]. Moreover, regular use increases dependence, which can lead to problematic use or cannabis use disorder (*CUD*) [24,25]. Moreover, among regular users, solitary cannabis use is more frequent, which is also related to coping motives and a higher prevalence of *CUD* symptoms [26,27].

Thus, the outbreak of the COVID-19 pandemic and the increase in psychological distress among Spanish university students [28,29] due to uncertainty and stress may have increased cannabis use, especially among users with regular or dependent use as a strategy to cope with the new situations [30].

Although some studies on the impact of the pandemic on substance use among university students have been developed in Spain [31,32], to our knowledge this is the first study that includes a ten-year time course that focuses on cannabis use and exploring the lockdown time and the new normal period.

The objectives of this study were to explore the impact of the COVID-19 pandemic on the prevalence and patterns of cannabis use among university students in contrast to previous trends since 2012, and secondarily to explore changes in other variables of interest such as places of cannabis use and age of onset.

## 2. Participants and Procedures

### 2.1. Study Design and Sample

A repeated cross-sectional study design was performed based on the basal survey of the uniHcos Project [33]. The study population was 11,491 university students from eleven Spanish universities (Universities of Alicante, Cantabria, Castilla—La Mancha, Granada, Huelva, Jaén, León, Salamanca, Valencia, Valladolid y Vigo) in the first year of any degree program who participated in the study filling the uniHcos baseline survey between January 2012 and May 2022.

The uniHcos questionnaire is an online self-administered survey based on the National Health Surveys of Spain and other validated questionnaires. The first version, which was administered from 2012 to 2019, consisted of 19 sections with a total of 453 assessable items, including other drug use such as alcohol and tobacco. After the outbreak of COVID-19, a section with 18 items to assess the impact of COVID-19 in different aspects was added.

Participants were recruited through their university account email. The information was collected through the online questionnaire on the SphinxOnline^®^ (v. 4.19, Le Sphinx Développement SARL, Annecy, France) digital platform, this platform kept data confidential and it complied with Spanish Law 3/2018 on Data Protection, in such a way that no researcher can find out the identity of the participants. The estimated response time for the questionnaire was between 30 and 45 min.

The only exclusion criteria for the analysis was the age of the participants, excluding participants over 25 years (*n* = 969; 8.4%), so the age range for the analyses was 17–24 years. This restriction was motivated by the interest of the study in the characterization of the young university population, together with the importance of having a sample as homogeneous as possible, as well as favor comparison with data from other surveys of the same age group. In total, a sample of 10,522 students (73.3% female) with mean age (standard deviation) of 19 (1.6) was included in the analysis. The sample size per wave on average (SD) was 1052(474). Since the uniHcos project is a dynamic cohort, no minimum sample size was established specifically for this study.

### 2.2. Study Variables

The sociodemographic variables considered were: sex, age, residence, alcohol and tobacco use, and people the student lived with. On the other hand, the prevalence of cannabis use was explored in three frequencies: ever in life, in the past 12 months, and in the past 30 days, assessed with the questions: “Have you used any of the following substances even once in your life?”, “How many days have you used cannabis in the past 12 months/30 days?”. These variables were categorized as dichotomous (YES/NO). Other variables on cannabis use were collected, such as frequent places of cannabis use and age of onset.

To assess possible changes in cannabis use patterns and cannabis use as a coping strategy, three questions were performed. The question about cannabis use during the first lockdown was: “During the first lockdown period, how did your cannabis use compare to before the lockdown?”, with possible answers being (1) “Cannabis use increased”, (2) “Cannabis use remained the same”, (3) “Cannabis use decreased”, (4) “I’m not a cannabis user”. The question about cannabis use during the COVID-19 pandemic was: “Since the arrival of the COVID-19 in Spain, has your cannabis use increased?”, with possible answers being (1) “Yes, it has increased”, (2) “No, the use is the same”, (3) “No, the use has decreased”, (4) “I’m not a cannabis user”. Finally, the question on coping as a reason for use was: “Since the arrival of the COVID-19 pandemic in Spain, have you frequently used substances such as cannabis to cope with the situation?”, with dichotomous answers YES/NO.

### 2.3. Data Analysis

A descriptive analysis was performed about the evolution over time of the prevalence of cannabis use in the three frequencies. For this objective, from 2012 to 2019, questionnaires were categorised by year of filling the survey. Since 2020, and for the purpose of assessing the possible impact of the COVID-19 pandemic, two periods of analysis were established: Lockdown (LD) and New Normal (NN). The lockdown period includes the questionnaires completed from May 2020 to May 2021 (no surveys were completed in the first months of 2020). This period was established in accordance with the Royal Decree (RD) 463/2020 of 14 March [34] and together with RD 926/2020 of 25 October [35] and its extensions, which establish the imposition of an “Alarm State” throughout the Spanish territory, implying severe mobility and social restrictions, since the derogation of “Alarm State” according to the RD-Law 8/2021 of 4 May [36]. The New Normal period includes from June 2021 to May 2022.

Pooled prevalence rates of cannabis use for each year and their 95% confidence interval were extracted. Sex-disaggregated analyses were also performed for the past 12 months and the past 30 days. A descriptive analysis of changes in cannabis use patterns during the first lockdown and during the pandemic among cannabis users, considering use for coping reasons, was conducted. The frequent places of cannabis use were assessed and the use in the usual household was also explored according to the people with whom the students lived. For these analyses, the data were categorized as: Before COVID-19 (BC), including from 2012 to 2019, LD, and NN, with the same criteria as before. Pearson’s Chi-square analyses were performed to assess significant differences between qualitative variables. To explore differences in the mean age of onset between sexes, the Mann–Whitney U test was performed. All statistical analyses were carried out with the software STATA 14.0 [37] with a significance level of 95% (*p*-value < 0.05).

### 2.4. Ethical Aspects

The study was carried out in accordance with the Code of Ethics of the World Medical Association (Declaration of Helsinki). All ethics committees of the participating universities evaluated and accepted this procedure. Participants collaborated voluntarily without any compensation and by completing an informed consent form accepting the conditions of the study.

## 3. Results

Table 1 shows the sociodemographic characteristics of the sample. The total sample size was 10,522, with 73.3% women. Most participants lived in a family household (46.1%) or a rented apartment (39.6%). Approximately half of the students lived with their parents (48.6%) while the other half lived with roommates or friends (47.6%). Tobacco and alcohol use prevalences by sex during the three periods of study are shown in Appendix A.

### 3.1. Evolution of Prevalence of Cannabis Use

Figure 1 shows the evolution of the prevalence of cannabis use in three frequencies (ever, past 12 months, and 30 past days) in each study period. As Figure 1 shows, the prevalence of cannabis use ever in a lifetime remained at similar values from 2013 to 2018, peaking in 2019 [48.3, (95%CI 42.6–54.0)]. Moreover, an increasing trend was observed from 2017 to 2019. In the lockdown period, this prevalence decreased to 44.0% (95%CI 40.1–47.9), showing similar values to previous years (2013–2018) and in the new normal period, this prevalence reached its minimum value [42.8%, (95%CI 38.8–46.9)] since 2012. Regarding the prevalence of cannabis use in the past 12 months, a slight upward trend was observed from 2012 to 2018, when it reached its highest value in the time series [33.7%, (95%CI 30.8–36.7)]. In the LD period, there is a notable decrease in prevalence from 32.1% (95%CI 27.0–37.6) in 2019 to 26.9% (95%CI 23.5–30.5); in the NN period, the prevalence of use reached again to similar values to those before LD [30.8%, (95%CI 27.1–34.6)]. Finally, a similar trend was observed in the past 30 days frequency.

The peak of this frequency was observed in 2013 [19.6%, (95%CI 14.4–19.79] the minimum value of the time series was observed in the LD period being 11.8% (95%CI 9.5–14.5). In this frequency of use in the NN period prevalence of use increased [14.5%, (95%CI 11.9–17.7)] but without reaching the previous values of the series.

Figure 2 shows sex-disaggregated data for the past 12 months and past 30 days prevalence of cannabis use. Prevalence in the past 12 months, by sex and according to Figure 2, was higher in males than in females from 2012 to LD. Showing statistically significant differences from 2013 to 2018 (*p* < 0.05 for chi-square test). In males this prevalence reached its highest value in 2016 [40.7%, (95%CI 35.2–46.4)], while in females it was in 2018 [31.3%, (95%CI 27.9–34.8)], being the same value as for the NN period [31.3%, (95%CI 27.1–35.8)].

In the LD period, a drop in the prevalence of use was observed in both sexes compared to previous years, being the prevalence of use similar between sexes (male: 28.5%, female: 26.4%). While in females the prevalence increased again in the NN period, in males it remained at a similar value to LD (29.2%).

Following the results of Figure 2, the prevalence in the past 30 days was higher in males than in females throughout the study period, showing statistically significant differences from 2014 to 2018 (*p* < 0.05 for the chi-square test), with a narrowing between the values for both sexes from 2019 to NN. In males, the highest value was observed in 2018 [23.7%, (95%CI 19.2–29.8)] a decrease in prevalence was observed in 2019, reaching its lowest value, 13.6%, in the LD period. In the NN period, it increased again, but without reaching before COVID-19 values. On the other hand, in females, the highest value was observed in 2013 [18.3%, (95%CI 16.0–20.9)]. As in males, there was a decrease in the prevalence of use in the LD period (11.2%), which increased in the NN period (13.6%) without reaching the previous values.

### 3.2. Changes in the Pattern of Use during the First Lockdown and during the COVID-19 Pandemic

Concerning the first lockdown (May-June 2020), 18.8% (95% CI 13.1–26.0), of cannabis users reported maintaining their use while 35.4% (95% CI 27.9–43.6) increased it and 45.8% (95%CI 37.7–54.1) decreased it. Regarding the pattern of use during the COVID-19 pandemic, Figure 3 shows the changes in the pattern of use during the pandemic, as well as the proportion of users who reported using cannabis as a coping strategy. According to our results, 20.3% (95%CI 15.4–26.3) of students reported a decrease in cannabis use during the pandemic, while 32.5% (95%CI 26.5–39.2) reported no changes in their use, and a large percentage of users, 47.2% (95%CI 40.5–53.8), increased it. About coping use, there were differences according to the pattern of use: while only 23.3% of those who decreased their cannabis use reported using for coping, this percentage was much higher among those who maintained their use (43.5%) and especially among those who increased it (53.0%). Furthermore, a statistically significant association (*p* < 0.05) was observed between increased cannabis use and its use as a coping strategy.

### 3.3. Frequent Locations of Cannabis Use

Among students who had used cannabis at least once in their lifetime and for all three study periods, Before COVID-19 (BC), Lockdown (LD), and New Normal (NN), private parties were the most frequent place of use, ranging from 47% to 55% between periods. University parties also played an important role, but their use in this place was greatly reduced during the lockdown period and increased again during the NN period, (BC: 19.2%, LD: 8.8%, NN: 15.6%). One of the most frequent places of use was the usual residence.

Figure 4 shows the cannabis use in the usual residence for the three periods according to whom the students lived with (parents or roommates/friends). As shown in Figure 4, for all three study periods, students living with roommates showed higher percentages of cannabis use in the usual residence than those living with their parents. A statistically significant association (*p* < 0.05) was found between living with roommates and higher consumption in the usual residence. It is worth noting the increase in cannabis in the usual residence during the LD period in both cases. For students living with their parents, use in the household increased from 16.9% (95%CI 19.3–28.7) BC to 26.3% (95%CI 19.5–34.3) in the LD and decreased again in the NN period to 11% (6.0–19.3). A similar pattern was observed among students living with roommates/friends, whose consumption at home increased from 36.4% (95%CI 34.4–38.4) BC to 43.2% (34.7–52.0) during the LD, decreasing to 35.3% (95%CI 27.7–43.6) in the NN period. Other less frequent places of use such as the street, bars, and the car were reported.

### 3.4. Age of Onset

The mean age of onset was similar across the time series, with a total mean age (standard deviation) of 16.6 years (1.7). No significant sex differences were observed (*p*-value = 0.7 for Mann–Witney U test). In addition, the percentile of students who started using between the ages of 14 and 18 was assessed. It was found that most students started using cannabis between the ages of 16 and 18, although more than 10% had already started using cannabis between the ages of 14 and 15. Since 2018, an earlier onset was observed, which stabilizes during LD and NN. These data are available in Appendix A.

## 4. Discussion

The main objectives of this study included the assessment of the evolution of the prevalence of cannabis use among first-year university students in Spain. According to our results, cannabis use ever in a lifetime showed an upward trend from 2017 to 2019, which otherwise was not reflected in an increase in past 30 day use, possibly indicating a higher number of students who decided to try cannabis but did not continue using it, in line with recent national trends [16].

The evolution of the prevalence of cannabis use from 2012 to 2019 showed values between 28–34% of students in the past 12 months, which are higher values than the national average for 15–24 years old, whose value was 22.1%. In the past 30 days, the prevalence rates were between 16–19%, which is like the national average of 15.9% [16]. Nevertheless, it was observed that our results on the prevalence of use were much higher than the European average (19.2% in the past 12 months, 10.3% in the past 30 days) for people aged 15–24 years [11], coinciding with other authors that Spain is one of the countries with the highest cannabis use in Europe [13].

About another objective of the study on the assessment of the potential impact of the COVID-19 pandemic, our results showed that during the lockdown period the prevalence of cannabis use decreased considerably, especially in the frequencies of the past 12 months (32.1% in 2019 to 26.9% in LD) and the past 30 days (16.2% in 2019 to 11.8% in LD). In the new normal period, these prevalence rates increased again. These results are consistent with other authors’ findings of lower cannabis use during COVID-19 restrictive periods in university students [38,39] and the general population [40,41]. This could be explained by the large number of social restrictions that influenced the reduction of mainly, occasional use, as other authors have suggested [38,39] as well as a possible lower availability [42,43] of the substance because of the mobility restrictions, especially during the first lockdown [11,23]. Likewise, in alcohol use, a decrease was observed during the lockdown in the past 30 days prevalence, with a reduction from 78.6% in the before COVID-19 period to 63.8% during the lockdown, recovering in the new normal period (77.7%) with no differences between sexes [44].

Disaggregating data by sex, it was observed that males reflected higher prevalence rates than females, although in the later years of the study these differences were not statistically significant. Although men have historically shown higher prevalence rates of cannabis use, in recent years it has been observed that women are increasing their prevalence rates, narrowing the cannabis use gap between men and women [45,46,47]. The analysis of the evolution in the next few years is a key point to shed light on this trend in university students.

On changes in patterns of use during the pandemic, it was found that among students who used cannabis, 47.2% increased their use since the beginning of the pandemic and 32.5% did not change their use. This represents a large percentage of students who increased or did not change their use (79.7%) compared to only 20.3% who decreased it. These results may be consistent with other authors’ findings of a reduction in occasional use but an increase or maintenance of use in many regular users in this period [48,49,50,51,52]. These patterns of use distribution may be related to an increase in dependence on the substance among regular users, so that despite social restrictions they continued using cannabis solitarily, which is common among regular users [27,53].

In line with this, it is worth noting our results on cannabis use as a coping strategy, as solitary cannabis use has been related to coping motives [53]. It was observed that among users who reported increasing their use, more than half reported using to cope with the pandemic, compared to 23.3% who reported decreasing their use. These data are of great interest since coping has been associated with an increase in depressive symptoms, increased risk of dependence, social vulnerability, and withdrawal symptoms [19,54,55,56], among other problems. Therefore, educating and managing coping strategies with students is of great interest to prevent problematic use and *CUD* among the university population.

Regarding places of use, parties were found to be among the most frequent places of use, reflecting the large number of students who consume in the social context. Use in the usual home was much higher among students living with roommates than among those living with their parents, which is consistent with other studies [18,57]. However, during the LD period, both students, living with roommates and living with their parents, increased their use. This contrasts with the reduction in cannabis use during LD that was observed in the evolution graphs reflecting a preference for consumption in the usual residence during the restrictive social measures. Moreover, this increase in use at home could reflect a high dependence on use coupled with a permissiveness of cannabis use in the home environment [58] or hiding it. The age of onset was similar across the time series, with no significant differences between the sexes. It was 16.6 years, which is lower than the national average (18.5) [16], but it must be considered that at the national level this data is calculated with a population aged 15–64 years.

The results obtained in this study should be interpreted with caution, as this study has some limitations. Firstly, the limitations of a cross-sectional study design make it difficult to provide causality, but as our main objective was to study the prevalence, we believe that the design is suitable for this purpose. Next, due to the uniHcos questionnaire aims to collect data on different lifestyle habits, variables about cannabis amount consumed or the form of cannabis use have not been collected. However, it has been established that among Spanish people, the most common form of use is marijuana as a joint [16]. In addition to this, data collection was obtained by a self-administered questionnaire not standardized, which could over- or underestimate the information obtained, especially considering the biases in terms of providing information on substance use and cannot be well generalized to other university populations or the entire population, although it could be considered a first step to understand trends that should be deepened in future research. Finally, there are also limitations in terms of the sample, as there was a higher percentage of women than men, which is expected to be controlled by stratification by sex, as well as differences in sample size in some periods.

On the other hand, it is interesting to highlight some of the strengths of our study, such as the possibility of understanding cannabis use in the Spanish university context after an analysis of its evolution over time and considering the influence of the pandemic, which provides new lines of research and new preventive strategies for this vulnerable population.

## 5. Conclusions

During the COVID-19 lockdown, the prevalence of cannabis use among university students decreased. Moreover, although males showed higher prevalence rates of use, in the last years, no significant differences between males and females were observed. Among the students who used cannabis during the pandemic, the majority maintained or increased its use. These results could imply that, within the university population, there are different patterns of cannabis use to deal with and implement policies both for social and regular use, contemplating coping use. Further study in the coming years is essential to assess how consumption trends evolve after the pandemic and to explore gender differences, as well as possible interactions with alcohol or tobacco use, and the use of other coping strategies, such as the use of social networks.

## Figures and Tables

**Figure 1 ijerph-19-11577-f001:**
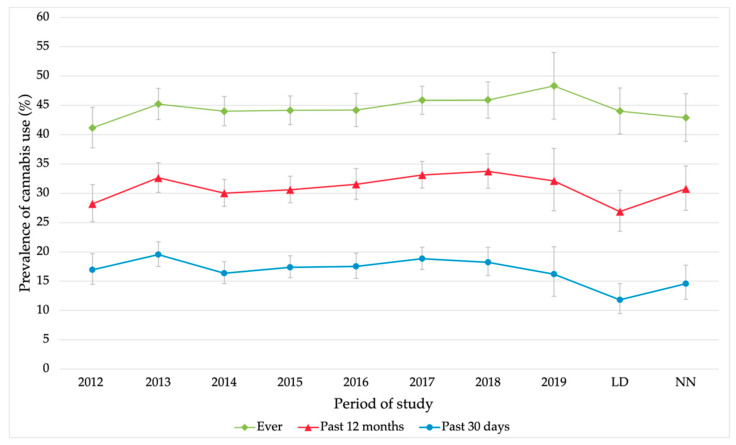
Evolution of the prevalence of cannabis use (%) in university students, ever, past 12 months and past 30 days. Error bars show the 95% confidence interval on the prevalence rates.

**Figure 2 ijerph-19-11577-f002:**
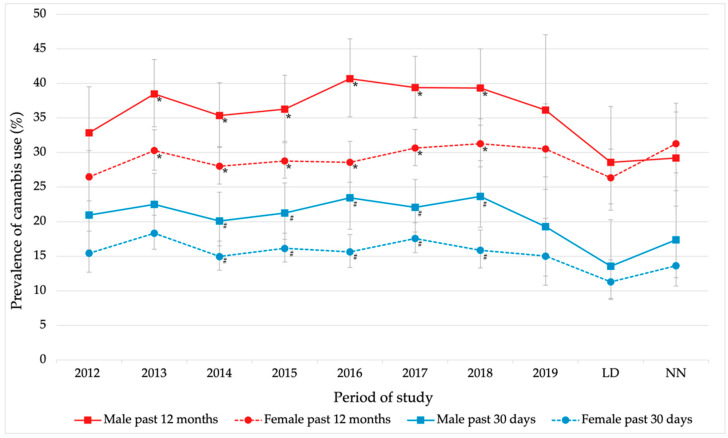
Evolution of the prevalence of cannabis use (%) in university students, past 12 months and past 30 days, by sex. Error bars show a 95% confidence interval on the prevalence rates. * *p* < 0.05 Male vs. Female (past 12 months); # *p* < 0.05 Male vs. Female (past 30 days).

**Figure 3 ijerph-19-11577-f003:**
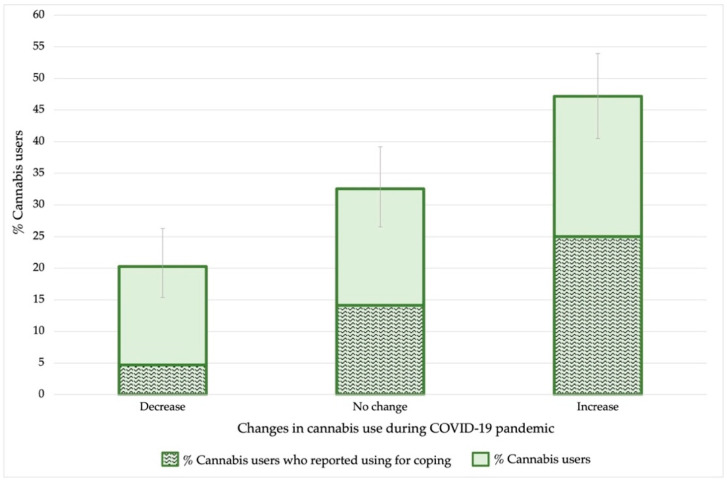
Changes in cannabis use during the COVID-19 pandemic and proportion of cannabis users who used cannabis for coping. Error bars show a 95% confidence interval on the proportion.

**Figure 4 ijerph-19-11577-f004:**
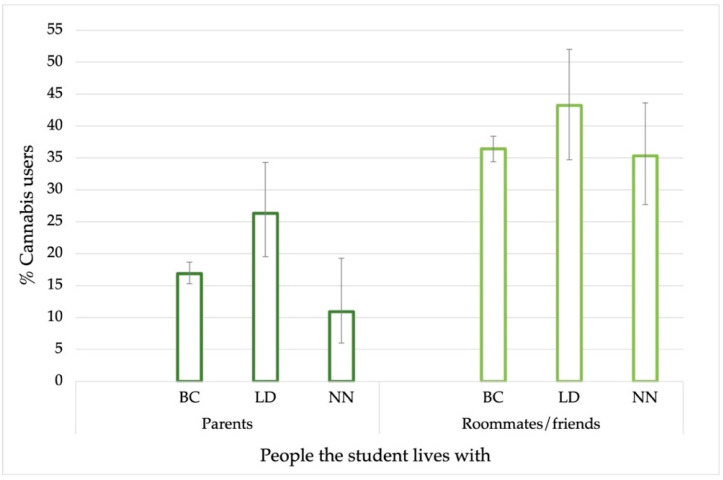
Cannabis use in the usual residence according to who the students lived with, during the three study periods (BC: Before COVID-19, LD: Lockdown, NN: New Normal). Error bars show a 95% confidence interval on the prevalence.

**Table 1 ijerph-19-11577-t001:** Sociodemographic characteristics of the sample.

	N	%
Total	10,522	100
Sex		
Female	7712	73.3
Male	2810	26.7
Period of study		
Before COVID-19 (BC)	9335	88.7
Lockdown (LD)	618	5.9
New Normal (NN)	569	5.4
Residence		
Family household	4847	46.1
Rented apartment	4171	39.6
University hall of residence	1297	12.3
Other	207	2
People the student lives with		
Parents	5113	48.6
Roommates/Friends	5013	47.6
Other	396	3.8

## Data Availability

The datasets analyzed in the current study are available from the corresponding author upon reasonable request.

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
