# Peer review of "Impact of the COVID-19 Pandemic on the Evolution of Prevalence and Patterns of Cannabis Use among First-Year University Students in Spain—UniHcos Project"

_ijerph, 2022, doi:10.3390/ijerph191811577_

Round 1

Reviewer 1 Report

The manuscript entitled "Impact of the COVID-19 pandemic on the evolution of prevalence and patterns of cannabis use among First-Year University Students in Spain. UniHcos Project" aimed to explore the evolution of the prevalence and pattern of cannabis use among university students and to explore the possible impact of the COVID-19 pandemic.

The study's theme is interesting and the manuscript is well written. I think it is almost ready for publication, but I think it can be improved in some points that I am going to point out below.

Abstract

It is not necessary to make a list by points (e.g., 1) .. 2) .. etc.), but it is necessary to follow a list that illustrates the theoretical background, the participants and the procedures, the results (briefly), and the conclusions. It is also desirable, in the participants’ section, to insert a brief description of the sample: the number of participating subjects, the mean and standard deviation of age, and the percentage of males or females.

Introduction

Although it is a very recent and well-articulated theoretical framework, I believe it is necessary to deepen the scientific literature on some points. It is also useful to introduce not only national but also international literature, specifically from similar countries of the study (Italy, France, Portugal, etc.). Here are some recent works that suit your theme and which I think may be useful for expanding and updating the introduction section.

About psychological distress, emotional stress, and negative emotions:

- Angelini, G., Buonomo, I., Benevene, P., Consiglio, P., Romano, L., & Fiorilli, C. (2021). The Burnout Assessment Tool (BAT): A contribution to Italian validation with teachers’. Sustainability13(16), 9065.

- López-Angulo, Y., Mella-Norambuena, J., Sáez-Delgado, F., Peñuelas, S. A. P., & González, O. U. R. (2022). Association between teachers’ resilience and emotional intelligence during the COVID-19 outbreak. Revista Latinoamericana de Psicología54, 51-59.

About the COVID pandemic and repercussions on mental well-being:

- Buonomo, I., Santoro, P. E., Benevene, P., Borrelli, I., Angelini, G., Fiorilli, C., ... & Moscato, U. (2022). Buffering the Effects of Burnout on Healthcare Professionals’ Health—The Mediating Role of Compassionate Relationships at Work in the COVID Era. International Journal of Environmental Research and Public Health19(15), 8966.

- Gori, A., Topino, E., & Di Fabio, A. (2020). The protective role of life satisfaction, coping strategies and defense mechanisms on perceived stress due to COVID-19 emergency: A chained mediation model. Plos one15(11), e0242402.

- Gori, A., & Topino, E. (2021). Across the COVID-19 waves; assessing temporal fluctuations in perceived stress, post-traumatic symptoms, worry, anxiety, and civic moral disengagement over one year of pandemic. International Journal of Environmental Research and Public Health18(11), 5651.

However, I invite you to deepen the literature on the subject to have a more articulated and complex overview of the variables studied.

Point 3: The aims of the study are not clear as there is no ad hoc section. Insert this section before the hypothesis section, noting that there must be a clear correspondence between the objectives and the results.

Method

Line 102: I would name this section "Participants and Procedures" since both are explained.

Lines 124-125: Better to make the section that refers to the mean and the standard deviation of age more explicit. Also better enter the gender percentages: e.g., "10522 students (Mage, 19 years; SD, 1.6; Male (or female), N%".

Discussion

Line 355: Within the limits, it is important to underline that this study refers to a series of information obtained from the participants that were not collected from standardized questionnaires. For this reason, the data cannot be generalized to the entire population. However, this can be a first step, a pilot study to be deepened in future research.

Author Response

We welcome suggestions, comments, and input from the editor and reviewers for improvement of the article.

We will then try to respond to the various suggestions and comments, and indicate whether any changes have been made to the document (highlighted in yellow in the manuscript).

Reviewer 1 suggestions and comments:

Com. 1.- Abstract

It is not necessary to make a list by points (e.g., 1) .. 2) .. etc.), but it is necessary to follow a list that illustrates the theoretical background, the participants and the procedures, the results (briefly), and the conclusions. It is also desirable, in the participants’ section, to insert a brief description of the sample: the number of participating subjects, the mean and standard deviation of age, and the percentage of males or females.

Answer 1: The abstract has been rewritten to remove unnecessary numbering, to include a better description of the sample, and to improve the organisation of the text, in accordance with the reviewer's suggestions. (Lines 41-56)

Com. 2.- Introduction

Although it is a very recent and well-articulated theoretical framework, I believe it is necessary to deepen the scientific literature on some points. It is also useful to introduce not only national but also international literature, specifically from similar countries of the study (Italy, France, Portugal, etc.). Here are some recent works that suit your theme and which I think may be useful for expanding and updating the introduction section.

About psychological distress, emotional stress, and negative emotions:

- Angelini, G., Buonomo, I., Benevene, P., Consiglio, P., Romano, L., & Fiorilli, C. (2021). The Burnout Assessment Tool (BAT): A contribution to Italian validation with teachers’. Sustainability, 13(16), 9065.

- López-Angulo, Y., Mella-Norambuena, J., Sáez-Delgado, F., Peñuelas, S. A. P., & González, O. U. R. (2022). Association between teachers’ resilience and emotional intelligence during the COVID-19 outbreak. Revista Latinoamericana de Psicología, 54, 51-59.

About the COVID pandemic and repercussions on mental well-being:

- Buonomo, I., Santoro, P. E., Benevene, P., Borrelli, I., Angelini, G., Fiorilli, C., ... & Moscato, U. (2022). Buffering the Effects of Burnout on Healthcare Professionals’ Health—The Mediating Role of Compassionate Relationships at Work in the COVID Era. International Journal of Environmental Research and Public Health, 19(15), 8966.

- Gori, A., Topino, E., & Di Fabio, A. (2020). The protective role of life satisfaction, coping strategies and defense mechanisms on perceived stress due to COVID-19 emergency: A chained mediation model. Plos one, 15(11), e0242402.

- Gori, A., & Topino, E. (2021). Across the COVID-19 waves; assessing temporal fluctuations in perceived stress, post-traumatic symptoms, worry, anxiety, and civic moral disengagement over one year of pandemic. International Journal of Environmental Research and Public Health, 18(11), 5651.

However, I invite you to deepen the literature on the subject to have a more articulated and complex overview of the variables studied.

Answer 2: Changes have been made in the introduction to delve deeper into the mental health aspects derived from the COVID pandemic both in Spain and in other countries by including more bibliographical references according to the reviewer's suggestions. (Lines 60-66; Ref. 2-4)

Com. 3: Point 3: The aims of the study are not clear as there is no ad hoc section. Insert this section before the hypothesis section, noting that there must be a clear correspondence between the objectives and the results.

Answer 3: The paragraph about aims has been rewritten to clarify correspondence with results. According to journal instructions for authors, the aims are described at the end of introduction section and not as a different ad-hoc section. (Lines 110-113)

Com. 4: Line 102: I would name this section "Participants and Procedures" since both are explained.

Answer 4: The section name has been changed according to the suggestion. (Line 115)

Com. 5: Lines 124-125: Better to make the section that refers to the mean and the standard deviation of age more explicit. Also better enter the gender percentages: e.g., "10522 students (Mage, 19 years; SD, 1.6; Male (or female), N%".

Answer 5: The information about sample characteristics has been included in the section (Lines 138-139)

Com. 6: Discussion

Line 355: Within the limits, it is important to underline that this study refers to a series of information obtained from the participants that were not collected from standardized questionnaires. For this reason, the data cannot be generalized to the entire population. However, this can be a first step, a pilot study to be deepened in future research.

Answer 6: The considerations on limitations and strengths of the study have been reworked to include the reviewer's comment on the limitations of using non-standardised questionnaires. (Lines 395-400)

Reviewer 2 Report

This manuscript presents data on cannabis use among Spanish students before and during different stages of the COVID pandemic. Data are based on cross-secional surveys gathered between 2012 and 2022.

The authors report interesting data on the evolution and conditions of cannabis use of male and female university students during this period. However, I think that the manuscript could be improved by adding more background information. Furthermore, if possible, a comparison with corresponding patterns of alcohol use would be important for identfying factors that are (un)specific for cannabis in comparison to another recreational drug. 

Comments / suggestions:

1. For international readers, it would add important information if the authors would include information on cannabis legislation in Spain. How does it differ from other EU countries? Has there been any change in the Spanish legislatio during the study period? If yes, could that have affected the results?

2. Does the survey the reported data are based on also include questions about the use of alcohol? If yes, it would be very interesting (and significantly increase the merits of this study) if the trajectory of cannabis use could be compared with the corresponding information regarding the use of alcohol.

Author Response

We welcome suggestions, comments, and input from the editor and reviewers for improvement of the article.

We will then try to respond to the various suggestions and comments, and indicate whether any changes have been made to the document (highlighted in yellow in the manuscript).

Reviewer 2 suggestions and comments:

Com. 1: For international readers, it would add important information if the authors would include information on cannabis legislation in Spain. How does it differ from other EU countries? Has there been any change in the Spanish legislation during the study period? If yes, could that have affected the results?

Answer 1: A sentence has been included on the current cannabis regulations in Spain and how they compare with those of other neighbouring countries. (Lines 64-80; Ref 14-15)

Com. 2: Does the survey the reported data are based on also include questions about the use of alcohol? If yes, it would be very interesting (and significantly increase the merits of this study) if the trajectory of cannabis use could be compared with the corresponding information regarding the use of alcohol

Answer 2: The survey conducted contains information on other lifestyle habits including alcohol, tobacco and other drug use.

We are aware of the potential interest in the joint analysis of data on different types of consumption, as well as their relationship with other variables related to mental health and coping. However, we believe that the information presented in this manuscript is highly relevant in itself, and requires a detailed presentation in an article focused on such data, and would be overly diluted in a more complex article including information from multiple consumptions.

A more detailed analysis and exposition of these interactions as well as possible protective or risk factors that are related to variations in behaviours would be left for further studies.

A comment on the need for such further studies has been included in the conclusions. Lines (410-419).

Round 2

Reviewer 2 Report

I agree with the authors that "a more detailed analysis and exposition of these interactions as well as possible protective or risk factors that are related to variations in behaviours would be left for further studies." However, I still think that including data as in Fig. 2 also with respect to alcohol consumption would greatly increase the merit of the article because this would allow to discriminate variations in cannabis consumption from variations in general drug consumption This must not be discussed in detail, but some possibility to discriminate between or generalize across these two drugs would be important.

Author Response

We thank the reviewer for his/her comments and suggestions:
In order to respond to the request to include information on alcohol use to compare cannabis with other substances:
-  We have included a small change in the methodology, where we have specified in more detail that we have collected information on alcohol use.(Line 126)
- The discussion has also been modified to include information on the variation in 30-day prevalence of alcohol consumption in the before-COVID-19, lockdown and new normal periods in comparison with what happens with cannabis. (Lines 350-353; Ref. 44).
All changes in this new version have been highlighted in green.